# Comparative Study of Intake, Apparent Digestibility and Energy and Nitrogen Uses in Sahelian and Majorera Dairy Goats Fed Hay of *Vigna unguiculata*

**DOI:** 10.3390/ani10050861

**Published:** 2020-05-15

**Authors:** Fafa Sow, Khady Niang, Younouss Camara, El Hadji Traoré, Nassim Moula, Jean François Cabaraux, Ayao Missohou, Jean-Luc Hornick

**Affiliations:** 1Department of Animal Resources Management and Nutrition Unit, Faculty of Veterinary Medicine, University of Liège, Quartier Vallée 2, Avenue de Cureghem 6, B43a, 4000 Liège, Belgium; sowvet2002@yahoo.fr; 2Senegalese Institute for Agricultural Research (ISRA), Bel Air, Routes des Hydrocarbures, BP 3120 Dakar, Senegal; dr.camara@yahoo.fr (Y.C.); elhtra@yahoo.fr (E.H.T.); 3Office of Zootechnie-Feeding, Inter-State School of Veterinary Science and Medicine (EISMV), BP 577 Dakar, Senegal; dyniang25@yahoo.fr (K.N.); missohou@gmail.com (A.M.); 4Department of Animal Resources Management, Faculty of Veterinary Medicine, University of Liège, FARAH Center, Quartier Vallée 2, Avenue de Cureghem 6, B43a, 4000 Liège, Belgium; Nassim.Moula@uliege.be (N.M.); jfcabaraux@uliege.be (J.F.C.)

**Keywords:** metabolism, digestibility, energy, nitrogen, breed, goat, *Vigna unguiculata*, variety 58/74

## Abstract

**Simple Summary:**

A specialized dairy goat like the Majorera (M) breed could be a good opportunity to meet the high demand for dairy products in Senegal. The efficiency of forage use from the M breed was therefore compared to that of the local Sahelian (S) breed. To this end, 6 M and 6 S goats were given a hay of a legume, *Vigna unguiculata*, as sole source of nutrient and their energy and nitrogen metabolism were studied. Forage intake was higher in the M goats but also their milk production. The efficiency with which energy and nitrogen was used was similar between the two breeds but M required more energy to meet its survival needs (energy for maintenance) and possibly less protein. Thus, it appears that M is probably better suited to dairy production systems requiring more attention from breeders.

**Abstract:**

This study aimed to compare digestive and metabolic characteristics in Sahelian (S) and Majorera (M) goat breeds. Six lactating females from each breed, with an average weight 27.0 ± 1.93 and 23.7 ± 1.27 kg, respectively, were used. Cowpea hay, variety 58/74, was offered as sole feed ingredient, at a rate of 2 kg of fresh matter per animal per day. The animals were placed in metabolic cages and a digestibility test was conducted according to an adaptation period of 15 days and a collection period of 7 days. The daily chemical components offered and refused and recovered faeces, urine and milk were measured in order to assess energy and nitrogen utilization. The M and S goats had similar levels of dry matter (DM) intake as well as nutrient digestibility. On a metabolic weight basis, dry matter intake, gross energy intake, metabolizable and energy intake, digestible energy and energy lost as methane production were significantly higher (*p* < 0.01) in M than in S goats. Urinary energy excretion was similar (*p* = 0.9) between breeds, while faecal energy output was higher in M than in S goats. The milk energy output from the M goats was higher than that the S goats (*p* < 0.05). However, metabolizable to net energy conversion efficiency (klm) was not affected by breed (*p* = 0.37), while N intake, milk N yield and faecal N losses, relative to metabolic weight, were significantly higher (*p* < 0.05) in M than in S goats. Similarly, the percentage of dietary N intake excreted in urine (UNIN) was higher in S than in M breeds. The breed factor had no effect on N retained, N digestibility, urinary N and N use efficiency. In conclusion, the M and S goats were similar in terms of energy and nitrogen use efficiency, despite higher daily milk production and DM consumption in the M goat. This suggests that the M breed is possibly more dependent on a dense nutrition diet than the S breed but requires less maintenance nitrogen.

## 1. Introduction

The feeding of livestock in tropical regions is a continuing problem that is currently taking on greater importance because of the desire of many countries, notably Senegal, to develop and improve their livestock breeding [1]. Establishment of a balanced ration requires prior knowledge of the nutritional requirements of animals and feed values of the various products and by-products used. In tropical Africa, the nutritional needs of domestic animals are not well understood. As for the feed values, they were up to now mainly extrapolated from the results of trials performed elsewhere [2].

In sub-Saharan Africa, goat supplementation is not a common practice among livestock farmers who give priority to cattle and sheep [3]. Yet, goats have withstood the Sahel drought better than other ruminant species [4]. This observation has led to renewed interest for this species in the last ten years, but the nutritional constraint remains a concern because it limits animal performance and above all weakens the organism against parasitic and infectious diseases [4]. Aboriginal ruminants in Sahel forage poor quality diets but are considered better adapted to valorise them when compared to their counterparts in temperate zones [4]. Thus, more precise knowledge of their nutritional metabolism would allow the conversion efficiency of diet into products to be improved [5].

Energy and protein are the two main components for feeding dairy goats and the most important factors in calculating diet [6]. Energy is the most common nutritional deficiency limiting productivity, while protein is an essential requirement for growth, pregnancy and milk production. Good pasture provides adequate protein for these needs [6]. Few studies [4,7] address the energy and protein metabolism that local breeds might have developed in the Sahelian (S) zone in response to their different needs, although their role in saving and securing agrarian systems among the poorest populations is still important, given their significant potential for meat and milk [8]. As Senegal is a country with an agricultural vocation, employing 60% of the rural population and with more than 750,000 family farms [9], there is justification for trying to increase livestock production by allowing goats to exploit available resources. Considering that, the ability to value protein and energy in local breeds is worth addressing [10]. Indeed, the development and changes in agriculture and livestock farming in the current agro-climatic, economic, sociological and political context of Senegal require the improvement of animal feed and the development of fodder crops in production systems [11]. Thus, the fodder cowpea (*Vigna unguiculata* (L.) WALP), one of the main species favored in this policy, allows, in addition to better animal feed for maintenance and traction, better production of quality milk and meat throughout the year, to improve soil fertility through the production of manure and the sequestration of carbon and nitrogen in the soil [12]. Cowpea is a leguminous, herbaceous plant of the papilionaceous subfamily, well known in Africa. It is a plant of warm regions with different environments and a temperature of 20 °C to 40 °C. It can be grown with between 300 and 1500 mm of rainfall and adapts from semi-arid to humid areas [13].

The importance of small dairy ruminants has increased significantly in recent years, especially in developing countries, where they are an interesting and important alternative for the supply of dairy products for human consumption [14]. The increased importance of dairy production is a consequence of its capacity to generate income, improving the living standards in rural subsistence-farming communities. It is therefore considered that dairy production in developing countries is an important tool to overcome social and economic issues, particularly child malnutrition and low-income generation [15]. Majorera (M) goats are characterized by their adaptation to semi-arid climates and by higher milk production with a mean production of more than 500 kg of milk per lactation (210 days) [16]. They account for about 70% of the goat population in the Canary Islands. Introducing M goats in Senegal was mainly motivated by adaptation characteristics similar to those of S breeds. The milk production of S goats, although relatively low in number per capita, is an important source of nutrients in the localities where they are raised [3]. Knowing that dry matter (DM) intake by ruminants varies according to size and genotype [17,18], the question should be raised whether feed intake and use of energy and protein may differ between breeds offered a nitrogen surplus diet consisting of a single ingredient such as cowpea, resulting in societal and utility impacts.

Thus, the aim of this study is to compare energy and protein metabolism between S and M goats fed legume hay forage cowpea, variety 58/74, in order to assess the effect of breed on DM intake and energy and nitrogen use efficiency.

## 2. Materials and Methods

### 2.1. Experimental Protocol, Animal and Plant Material

In the absence of proper regulation on the use of animals for research and animal welfare during experiments in Senegal, the protocols were carried out according to the best practices usually accepted by the Ethical Committee of the University of Liège (Belgium) when conducting similar experiments.

The experiment was conducted at the Sangalkam research station (Latitude 14°46′ 44,30″ North, Longitude 17°13′ 33,65″ West, Altitude 19 m), in the department of Rufisque located in the Niayes ecological zone of the Dakar region, Senegal. The climate is sub-Saharan and the soils are sandy-clayey and rich in organic matter (OM). In the hot and rainy season (June to October), average temperatures range from 25 to 30 °C with an average rainfall of 400–500 mm. In the cool season (November to April), average temperatures vary between 19 and 23 °C. The average annual rainfall is about 400 mm. The test was carried out from February 2 to 24, 2018, and included two steps: a 15-day adaptation, followed by a 7-day measurement. A total of 12 healthy females, including 6 S and 6 M were used. All animals were mature and lactating with an average weight (mean ± SD) of 27.0 ± 1.93 and 23.7 ± 1.27 kg, respectively in S and M breeds. They were 5 to 7 years old, 3 to 5 lactation ranks and 8 to 9 weeks of lactation stage. The initial daily milk production was 0.24 and 0.76 L/d in S and M groups, [3,19] respectively. One week before the adaptation period, all animals were dewormed with Ivermectin^®^ 1%, received vitamins (stress vitam^®^) and were vaccinated against plague of small ruminants, smallpox, pasteurellosis and enterotoxaemia. They were housed in individual stalls (dimensions 190 × 65 cm) with feeder and water devices. The diet consisted of 100% haulm of cowpea (*Vigna unguiculata* (L.) Walp, variety 58/74; Table 1), a legume widely cultivated in sub-Saharan Africa [20]. Fodder was harvested from legumes grown between mid-August and mid-October 2017 in Sangalkam station, during the rainy season, preceding the experimental period. The whole plant was harvested 8 weeks after sowing and air-dried in the shade. Daily, 2 kg of cowpea haulm per animal was chopped and divided into two meals of equal size offered at 8:30 am and 2:30 pm. The animals had free access to water and did not receive any mineral supplements. The daily feedstuff intakes were calculated by difference between the amounts offered and refused.

### 2.2. Sample Collection and Chemical Analyses

During the seven days of experimentation, 100 g feed samples were daily collected and pooled. Refusals and faeces were weighed each morning before a new ration was distributed, and samples (100 g) were placed in labelled plastic bags, allowing individual identification of the animals, and stored at −20 °C. Composite samples of the collected material were dried at 60 °C to constant weight in a ventilated oven, ground to pass a 1-mm screen (Wiley mill, Marconi, MA-580; Piracicaba, Brazil) and stored in sealed plastic containers for subsequent analyses. A daily sample of 5 mL of urine per urination, with an average of 5 urinations per goat per day, was collected between 8.30 a.m. and 6 p.m. and then placed in single vial and immediately stored at −4 °C to avoid nitrogen volatilization until chemical analysis. Samples (175 mL) of urine were collected for nitrogen, energy and creatinine determinations from individual pooled samples [21]. The average creatinine excretion of 0.197 mmol/kg body weight, previously determined by total urine collection in adult Hereford cows [22], was used to calculate urine volume as: urine volume (L) = (0.197 (mmol/kg body weight) × body weight (kg))/creatinine excretion (mmol). This choice is supported by [23,24], which according to their studies in ruminants (cattle, buffaloes, sheep and goats) and rabbits, urinary creatinine concentration is affected neither by diet nor by the physiological state of the animal but is excreted in proportion to the body weight of any of the species studied. To ensure the absence of animal weight change during the study, the goats were weighed each morning after complete milking. Milk production was registered, and a 10 mL sample was collected in vials and stored at −4 °C for further analysis. All samples were analyzed for DM, ashes and crude protein (CP) using AOAC International methods [25]. For feed and faeces, the contents of neutral detergent fiber (NDF), acid detergent of lignin (ADL) and acid detergent fiber (ADF) were determined using the Van Soest method [21]. Gross energy of the composite samples was determined using an adiabatic bomb calorimeter (Werke C2000, IKA; Staufen, Germany).

### 2.3. Calculations and Statistical Analysis

Apparent nutrient digestibility were determined from feed intakes and faecal losses [26]. Nitrogen and energy balances were obtained by taking into account energy and nitrogen losses in urine, faeces and milk, and estimation of energy lost in methane [27]. Per animal daily average gross energy (GE) intake, faecal energy lost (FE), digestible energy (DE) intake, energy lost in urine (UE), energy lost as methane (ECH4), metabolizable energy (ME) intake, energy excreted in milk (NEl) and milk efficiency ratio (klm) were deducted from the 7-day measurement period and determined as follow [28,29]:GE (kcal/animal//d) = gross energy feed (kcal/kgDM) × DM intake (kg/animal/d)(1)
UE (kcal/animal//d) = gross energy urine (kcal/kgDM) × urine DM (kg/animal/d)(2)
FE (kcal/animal//d) = gross energy faeces (kcal/kgDM) × faecal DM (kg/animal/d)(3)
NEl (kcal/animal//d) = gross energy milk (kcal/kgDM) × milk DM (kg/animal/d)(4)
DE (kcal/animal//d) = GE − FE(5)
ME (kcal/animal//d) = DE − ECH4 − UE(6)

ECH4 was estimated from the equation of [23,24,25,26,27,28,29,30], using ECH4%GE = 3.67 + 0.062 × dE, where ECH4%GE is the energy lost in CH4 in kcal/100kcal GE, dE the apparent energy digestibility coefficient, and ECH4 (Kcal/animal//day) was obtained as GE ×% CH4.
q = (GE – FE – UE − ECH4)/GE = ME/GE(7)
and klm was estimated from 0.65 + 0.247 × (q − 0.63), INRA, 2018 [31].

Similarly, daily average nitrogen intake (Ni), urinary nitrogen (UN), faecal nitrogen (FN), milk nitrogen (MN), N digestibility (ND), nitrogen retained (NR), nitrogen utilization efficiency (NuE) and dietary N intake excreted in urine (UNIN) were determined as follows:Ni (g/animal/d) = crude protein intake (CP, g/animal/d)/6.25(8)
FN (g/animal/d) = faecal crude protein (FCP, g/animal/d)/6.25(9)
MN (g/animal/d) = milk protein (MP, g/animal/d)/6.38(10)
whereby, according to the Kjeldahl method, the coefficient of 6.38 allows the transformation of the determined amount of nitrogen into protein weight;
UN (g/animal/d) = N-NH4 (g/L) × urine volume (L/d)(11)
where N-NH4 is ammoniacal nitrogen;

Apparent N digestibility (Nd) was determined from feed N intake and faecal N loss.
NR (g/animal/d) = Ni − (FN + UN);(12)
NuE (%) = NR/Ni.(13)
UNNi (%) = UN/Ni.(14)

For the purpose of energy and nitrogen metabolism description, the results were expressed per kg of metabolic body weight.

Data on intake, digestibility, energy and urinary metabolism parameters were analyzed as means at group level using the mixed procedure PROC ANOVA from SAS/STAT^®^ software, SAS System for Windows Version 9.4 (SAS Institute Inc. Cary, NC, USA) according to the following statistical model: Yi = μ + αi + ei(15)
where Yi is the dependent variable, μ is the overall mean, αi is the fixed effect of breed (i = 1,2), and ei is the residual error. The F-values were considered significant at *p* < 0.05 and tendencies were assumed at 0.05 < *p* ≤ 0.10. Data reported are LS means and standard error of mean.

## 3. Results

### 3.1. Intake and Apparent Digestibility

Results from DM, chemical component intake and apparent digestibility are shown in Table 2. Expressed on a live weight basis, the feed DM intake tended to be lower (*p* < 0.09) in S than in M goats. When regard to metabolic weight, the value was significantly higher (*p* < 0.05) in M than in S goats. Although no significant differences were observed between the two breeds for DM and OM digestibility, however near trends for differences could be noted (*p* = 0.1).

### 3.2. Energy and Nitrogen Uses

Results on weight parameters and energy and nitrogen uses are shown in Table 3.

The live or metabolic weight of M was significantly lower (*p* < 0.003) than that of S goats. Gross, digestible and metabolizable energy intake (Kcal/kg^0.75^) were significantly higher (*p* < 0.01) for M than for S goats. While UE output was similar (*p* = 0.9) between the two breeds, the amount of energy lost in the faeces was higher (*p* = 0.01) for M than for S goats. Similarly, net milk energy output was higher (*p* < 0.05) for M than for S goats, with more than doubled values in M, when compared to S. Some effects between breeds were observed on energy use efficiency. Therefore, the lower values of DE observed in S goats were compensated by lower energy losses in methane and, as a result, resulted in similar klm values (*p* = 0.37). Energy digestibility was close to 59% and similar between the two races. The UE/DE ratio was numerically lower in M than in S (10.6 vs. 15.0, respectively, *p* = 0.17), while the ECH4 was similar at 6.25%. The ME/DE ratio was numerically higher in M than in S (due to the lower proportion of UE loss). Consequently, the q-value was lower in S, but the calculated klm values were close and of the order of 62%.

Nitrogen intake and faecal N relative to metabolic weight were significantly higher (*p* < 0.01) for M than for S goat. Similarly, milk N was 41% higher (*p* < 0.05) for M than for S goats. The percentage of dietary N intake excreted in urine (UNIN) accounted for about 60% and was higher in S (*p* < 0.05) than in M. No significant differences (*p* > 0.1) were observed between the two breeds in UN output and N retained, and N use efficiency; however, near trends for differences in N digestibility could be noted (*p* = 0.1). The percentage of N intake recovered in milk was doubled in the M group, when compared to Sahelian animals (7.17 vs. 3.6%, *p* < 0.04).

## 4. Discussion

### 4.1. Intake and Apparent Digestibility of Nutrients

The natural feeding behavior of goat is not negligible, as according to [32] goats are more selective ruminants. They preferably feed on the most nutritious parts of plants [33,34]. However, in this experiment, the animals had access to high quality fodder and few refusals were observed. It is noteworthy that the fodder contained about 15% CP in DM, indicating the high nutritional value of the feedstuff [35] and adequate availability of fermentable nitrogen in rumen. Our results are lower than those [27,36] in dairy goats fed alfalfa, hay and concentrate or ruminants (goats, sheep and llamas) fed a green hay and straw-based ration.

The results on the DM intake in relation to metabolic weight showed a greater ingestion capacity of M than S. According to [37], it is difficult to compare feed consumption values, as they are known to be influenced by factors such as the characteristics of the animals, environmental conditions, management type and their interactions. In the context of this experiment, animal factors only differed between the two groups and should be considered. The higher metabolic DM intake in M goats is probably associated to the higher milk production observed in this group (0.37 ± 0.2 vs. 0.15 ± 0.07 L). Considering that in goats DM intake increase by 0.405 kg per liter of milk production [31], the corrected DM intake for maintenance were close at 72 and 74 g per kg metabolic weight in M and S breeds, respectively. This is considered as close to the standard value obtained with good forage. Our results are similar to those of [4], who found an ingested DM of 64 and 93 g/kg^0.75^ in lactating S goats.

As for DM and nutrient digestibility, our results are lower than those reported by [27,36], except for the digestibility of NDF, which showed close values. There was no difference between the two breeds on digestibility of all the chemical constituents [38]. However, trends in the differences in digestibility of DM and OM to the advantage of S goats could be observed (*p* = 0.1). Chewing and rumination time, salivation and rumen motility are among the factors influencing digestibility [39], and the higher the intake, the lower the digestibility, which is explained by an increase in rumen transit time [40].

### 4.2. Energy and Nitrogen Uses

Despite highly significant differences (*p* < 0.05) in favor of M goats on faecal energy and methane and a corresponding very high significant difference (*p* < 0.01) in metabolizable energy (ME) intake, the different energy transformation coefficients showed especially a higher proportion of energy lost as urine in S goats. This is probably due to lower protein catabolism associated to higher milk production in the M group. However, the calculated klm values were similar at 0.6 for M and S goats. Our results on energy efficiency were lower than those found by [18,41] (0.67 and 0.695, respectively). According to [29], under tropical conditions, ruminants are generally fed low-protein forages with high NDF content and low DM digestibility and ME concentration. These factors result in lower use of ME as more energy is lost as heat due to effort of DM digestion. In the context of this experiment, the feedstuff offered could be considered, with close to 70% of GE being digestible, as good quality, which probable reduced this phenomenon. Depending on the breed, our results showed a superiority (*p* < 0.01, Table 3) in favor of the M over the S goats with regard to metabolizable energy intake (EM, kcal/day/kg^0.75^) with M showing 35% higher values than the S goats. Assuming a similar value of klm, and considering the net energy (NE) of lactation obtained in the two breeds of this experiment, the respective maintenance ME intakes per kg metabolic weight were 151 and 115 kcal, i.e., 32% higher in M when compared to S. These results corroborate those of [18], who found the metabolizable energy for maintenance 8% higher for dairy goats compared to local goats. According to [17], several factors influence maintenance energy requirements, such as breed, sex, age, environmental conditions and activity. The ME and probably also the NE values for maintenance of S goats are lower than those obtained in exotic breeds such as Saanen and Boer, which are more specialized in milk or meat production [18]. Our results corroborate those found by [33], while they are similar to those found by [27].

Nitrogen intake and faecal and milk nitrogen relative to metabolic weight were significantly higher (*p* < 0.05; Table 3) in the M than the S goats, but urine nitrogen outputs were very similar. The numerically higher N digestibility in the S group is probably of little importance, if any, and it could be associated with lower feed and N intake in this group. According to [41], due to the mechanisms of N homeostasis, faecal and urinary nitrogen excretion are closely linked to N intake in ruminants. However, the similar urinary N observed in this experiment suggests that N was better converted into milk N in the M group. Indeed, M animals were twice as efficient at transferring N to the milk. This phenomenon could be due to lower protein maintenance requirement, but this needs to be confirmed. For both breeds, nitrogen retained was close to but less than zero (i.e., greater amounts of nitrogen were excreted in the faeces and urine than those ingested with the diet). This suggests that although the weight of the animals remained constant during the experiment, body compartment changes could occur and that the conclusions of this study have to be considered carefully.

## 5. Conclusions

Cowpea hay fodder, var.58/74, appears to be a high-quality fodder for goat. When M breed goats were offered this fodder, they showed a greater flux of energy and nitrogen across the organism than S goats, but efficiency of energy use was quite similar, despite a higher basal metabolism rate in the European breed. In hard conditions, the S breed is probably better adapted to survive. M goat is more productive but requires probably higher quality feedstuffs and more attention from the breeder. However, further studies are still needed to confirm these findings, especially the paradoxically lower nitrogen maintenance requirement in M goats.

## Figures and Tables

**Table 1 animals-10-00861-t001:** Chemical composition of Cowpea fodder haulm (*Vigna unguiculata*), (Cowpea, var.58/74) used in the experiment.

OM	CP	EE	CF	ADF	NDF	ADL	GE (kcal/kg DM)
(% DM)							
76.9	15.0	2.60	11.4	38.6	55.3	5.40	3996

% DM = Percentage of dry matter; OM = organic matter; CP = crude protein; EE = ether extract; CF = crude fiber, ADF = acid detergent fiber; NDF = neutral detergent fiber; ADL = acid detergent of lignin; GE = gross energy.

**Table 2 animals-10-00861-t002:** Dry matter, chemical component intake and apparent digestibility (%) in lactating goats from Majorera and Sahelian breeds fed cowpea fodder.

Goat Breed				
Chemical Composition	Majorera	Sahelian	SEM	*p*-Value
Intake (kg/animal/d)				
DM	1.0	0.86	0.05	0.09
DM (g/kg^0.75^)	92.7	70.2	5.22	0.01
OM	0.87	0.76	0.04	0.13
EE	0.04	0.03	0.002	0.09
CP	0.18	0.15	0.01	0.07
NDF	0.46	0.39	0.03	0.13
ADF	0.27	0.20	0.03	0.14
Apparent digestibility (%)				
DM	58.5	64.5	2.41	0.1
OM	59.9	64.9	2.02	0.1
EE	61.3	71.6	4.79	0.15
CP	54.3	61.4	3.12	0.13
NDF	55.1	58.9	3.20	0.42
ADF	46.9	39.9	5.74	0.41

The significance effect were considered at *p* < 0.05; SEM: standard error of means; DM: dry matter; OM: organic matter; EE: ether extract; CP: crude protein; NDF: neutral detergent fiber; ADF: acid detergent fiber.

**Table 3 animals-10-00861-t003:** Weight parameters, energy and nitrogen uses in lactating goats fed cowpea hay var.58/74.

Goat Breed				
Parameters	Majorera	Sahelian	SEM	*p*-Value
Live weight (LW.kg)	23.7	27.0	0.59	0.003
Metabolic weight (kg^0.75^)	10.8	11.9	0.19	0.003
Milk production (L/d)	0.37	0.15	0.06	0.02
**Energy metabolism** (kcal/d/kg^0.75^ or%)				
Gross energy intake	372.8	268.3	22.44	0.008
Faecal energy	154.2	109.7	10.43	0.01
Digestible energy intake	218.6	158.6	12.75	0.007
Energy digestibility (%)	58.6	59.2	0.98	0.68
Urinary energy	23.2	23.8	3.98	0.9
UE/DE (%)	10.8	15.6	0.02	0.17
ECH4	13.8	9.9	0.83	0.008
ECH4/DE (%)	6.2	6.3	0.001	0.5
Metabolizable energy intake	195.4	134.8	13.28	0.009
ME/DE (%)	82.9	77.9	2.43	0.17
q = ME/GE (%)	48.6	46.3	1.8	0.37
Net Energy of lactation	27.5	12.5	4.39	0.04
klm	62.4	61.8	0.04	0.37
**Nitrogen metabolism** (g/d/kg^0.75^ or%)				
Nitrogen intake	2.6	2.1	0.1	0.004
Faecal nitrogen	1.2	0.8	0.1	0.01
Urinary nitrogen	1.46	1.44	0.05	0.78
Milk nitrogen	0.2	0.07	0.03	0.02
N digestibility (%)	54.3	62.4	3.4	0.10
UNIN (%)	55.2	69.7	4.2	0.04
Nitrogen retained	−0.01	−0.13	0.12	0.51
NuE (%)	−0.90	−7.32	5.58	0.44

The significance effect were considered at *p* < 0.05; SEM: standard error of means; UE: urinary energy, DE: digestible energy; ECH4: energy lost in methane; ME: metabolisable energy; GE: gross energy; klm: efficiency of energy use for lactation and maintenance; UNIN: percentage of dietary N intake excreted in urine; NuE: nitrogen utilization efficiency.

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
