# Peer review of "Comparative Study of Intake, Apparent Digestibility and Energy and Nitrogen Uses in Sahelian and Majorera Dairy Goats Fed Hay of Vigna unguiculata"

_animals, 2020, doi:10.3390/ani10050861_

Round 1

Reviewer 1 Report

Ms. Ref. No.:  animals-766179

Title: “Comparative study of intake, apparent digestibility and energy and nitrogen uses in Sahelian and Majorera dairy goats fed hay of Vigna unguiculata, variety 58/74”

Animals

General comments

I have had the opportunity to review the manuscript. The manuscript is interesting for this geographical are, the part related to feed goat and nutrition, but more details are necessary for the part related introduction, material and methods and in particular needed to add more references to support and explain this study.

Below my considerations:

Title

I suggest to delated in the title the number of variety “variety 58/74”, I suggest to add this in Keywords

Abstract

L32. The 2 kg must be indicated if they are dry matter or not

Introduction

The introduction is poor on the back ground of the study. It is necessary to expand the knowledge of the two breeds of goats under study and because what has been written and affirmed in several points, it is necessary to add the bibliographical references.

L 51-53 Add the references

L 57-58 Add the references

L 61-63 Add the references

L 66-67 Add the references

L 67-69 Add the references

L 69 “few studies” indicate what these studies are, add the references

Material and methods

L 94-95 You need to add the number of lactations and the average daily milk production.

L 100 The reference indicated is not clear

L103 “2 kg” indicate whether it is dry matter or not

L107 “Table 1” I suggest to deled and show the chemical composition inside the text

L129-133 This part appears to be confused, please reformulate

L135 Add the references

L 136-137 Add the references

L 142-163 all equations must have precise references, I suggest to group all equations in a table

L164-167 The statistical analysis model is not correctly described. It is necessary to indicate the factor and the variables analyzed, indicate the level of accepted significance

Results

L 173-175 The sentence is not clear. If there are no significant differences, it is enough to say it.

L 176 “Table 2” deleted “var.58/74”; replace “Items” with “Chemical composition”

L 182-184  The sentence is not clear, reformulate and add reference

L 187-188 Rewrite or delete

L 194 “P>0.1" it is not necessary, delete

Discussion

L 210 If you talking about gallons of milk, they should be shown with a standard average and a standard error. Report the values

L 210 “In S goats, the value…”  Specify to which value it refers

L 211-212 Add the references

L 214-215 Unclear statement, rephrase add the following citation https://doi.org/10.1016/j.smallrumres.2014.10.008 and rephrase the discourse

L 235; 252; 256. It is incorrect to enter a quote referring to another quote, each quote must be added if it wants to be in the comment, correct

L 257-259 It is not correct to say that other authors say “…for the same authors, our results are lower….” please reformulate 

References

All the bibliography must be revised and corrected according to the guidelines of the journal.

Author Response

General comments

I have had the opportunity to review the manuscript. The manuscript is interesting for this geographical are, the part related to feed goat and nutrition, but more details are necessary for the part related introduction, material and methods and in particular needed to add more references to support and explain this study.

 => Ok this has been done

Below my considerations:

 Title

I suggest to delated in the title the number of variety “variety 58/74”, I suggest to add this in Keywords

 => Ok this has been done (L1-4) and (L48)

Abstract

L32. The 2 kg must be indicated if they are dry matter or not

 => Ok this has been done (L31)

Introduction

The introduction is poor on the back ground of the study. It is necessary to expand the knowledge of the two breeds of goats under study and because what has been written and affirmed in several points, it is necessary to add the bibliographical references.

L 51-53 Add the references 

=> Ok this has been done (L53)

L 57-58 Add the references

=> Ok this has been done (L58)

L 61-63 Add the references

 => Ok this has been done (L61; L63)

L 66-67 Add the references

=> Ok this has been done (L 67)

L 67-69 Add the references

=> Ok this has been done (L67; L69)

L 69 “few studies” indicate what these studies are, add the references

 => Ok this has been done (L 69)

Material and methods

L 94-95 You need to add the number of lactations and the average daily milk production.

=> Ok this has been done (L120 – 121)

L 100 The reference indicated is not clear

=> Ok this has been done (L 127)

L103 “2 kg” indicate whether it is dry matter or not

=> Ok, the  “2 kg” indicate fresh matter (L 129 – 130)

L107 “Table 1” I suggest to deled and show the chemical composition inside the text

 => Ok, thank you for the recommendation, but we think the table is more clear (L 133– 134)

L129-133 This part appears to be confused, please reformulate

 => Ok this has been done (L 157 – 162)

L135 Add the references  

=> Ok this has been done (L170)

L 136-137 Add the references 

 => Ok this has been done (L 170)

L 142-163 all equations must have precise references, I suggest to group all equations in a table

=> Ok, thank you for this comment, but we think that they are better explained in this presentation, because they are mostly classical calculations referring to many authors (L 170 – 193)

L164-167 The statistical analysis model is not correctly described. It is necessary to indicate the factor and the variables analyzed, indicate the level of accepted significance

=> Ok this has been done (L 196 – 202)

 Results

L 173-175 The sentence is not clear. If there are no significant differences, it is enough to say it.

=> Ok, thank you. In our opinion, a value of P = 0.1 is generally worthy to be indicated, as currently observed in publication data. (L 209 – 211)

L 176 “Table 2” deleted “var.58/74”; replace “Items” with “Chemical composition”

=> Ok this has been done (L 239)

L 182-184  The sentence is not clear, reformulate and add reference

=> Ok this has been done (L 219 – 221)

L 187-188 Rewrite or delete

=> Ok this has been done (L 224)

L 194 “P>0.1" it is not necessary, delete

=> Ok this has been done (L 225 – 226)

 Discussion

L 210 If you talking about gallons of milk, they should be shown with a standard average and a standard error. Report the values

=> Ok this has been done (L 259 – 260)

L 210 “In S goats, the value…”  Specify to which value it refers

=> Ok this has been reformulated (L 260 – 262)

L 211-212 Add the references

=> Ok this has been done (L 260)

L 214-215 Unclear statement, rephrase add the following citation https://doi.org/10.1016/j.smallrumres.2014.10.008 and rephrase the discourse

=> Ok this has been done (L 264)

L 235; 252; 256. It is incorrect to enter a quote referring to another quote, each quote must be added if it wants to be in the comment, correct

=> Ok this has been done (L 247 – 278)

L 257-259 It is not correct to say that other authors say “…for the same authors, our results are lower….” please reformulate 

=> Ok this has been done (L 264 – 265)

References

All the bibliography must be revised and corrected according to the guidelines of the journal.

=> Ok this has been done (L 323 – 430)

Reviewer 2 Report

The manuscript has good potential but as is, it lacks the high quality that an international scientific journal requires. Starting from the introduction, authors need to justify in details the objective of the study, specifying why they picked these two breeds and they fed that specific forage. Regarding the design and the stats, some of the results will depend on the initial values (i.e. a covariate will need to be included in the analysis and that will possibly result in different conclusions). The number of animals is also limited so a power test could also make the study more robust. Most of the results are also expected since 1 of the breeds has been more selected. Most of the european breeds will make more milk so again such a comparison need to be justified to avoid that studies like this do not simply justify the importation of european breeds, since it is well known what the consequences have been historically. Also, the manuscript needs to be revised by a person proficient in English and, possibly, in the topic. Below are only some suggestions. A better use of appropriate references is needed as well.

Line 41: breed

Line 55: are

Line 58-59: a reference is needed

Line 75: resources

Line 75: period not needed?

Line 78: need to expand on cowpea and the specific breeds in the introduction, which is too broad. Justify the objective better.

Line 93: fix English

Line 95: indicate age even if mature animals are used and initial milk yield Line 95: indicate days since kidding

Line 104: specify the form the cowpea was offered (whole? chopped?...)

Line 107/Table 1: CF can be removed

Line 110: DM as %DM is 100! No need to show DM

Line 111: CA is not shown (‘Ash’ is enough)

Line 114: were

Line 115: removed and weighed

Line 118: need to indicate times of day? consistent times?

Line 121: need references

Line 123: justify use of cows’ urine volume

Line 160: loss

Line 166: what about use of co-variates?

Line 182: are these final weights?

Line 184: breeds

Line 204: not based on live weight

Line 216: motility

Line 218: based on what?

Author Response

Animals

The manuscript has good potential but as is, it lacks the high quality that an international scientific journal requires. Starting from the introduction, authors need to justify in details the objective of the study, specifying why they picked these two breeds and they fed that specific forage. Regarding the design and the stats, some of the results will depend on the initial values (i.e. a covariate will need to be included in the analysis and that will possibly result in different conclusions). The number of animals is also limited so a power test could also make the study more robust. Most of the results are also expected since 1 of the breeds has been more selected. Most of the european breeds will make more milk so again such a comparison need to be justified to avoid that studies like this do not simply justify the importation of european breeds, since it is well known what the consequences have been historically. Also, the manuscript needs to be revised by a person proficient in English and, possibly, in the topic. Below are only some suggestions. A better use of appropriate references is needed as well.

=> Ok this has been done

Line 41: breed 

=> Ok this has been done (L 40).

Line 55: are

=> Ok this has been done (L 55).

Line 58-59: a reference is needed

=> Ok this has been done (L 58 – 59).

Line 75: resources

=> Ok this has been done (L 75).

Line 75: period not needed?

=> Ok this has been done (L 75).

Line 78: need to expand on cowpea and the specific breeds in the introduction, which is too broad. Justify the objective better.

=> Ok this has been done (L 79 – 85) and (L 92 – 97).

Line 93: fix English

=> Ok this has been done (L 117 – 118).

Line 95: indicate age even if mature animals are used and initial milk yield Line 95: indicate days since kidding

=> Ok this has been done (L 119 – 122).

Line 104: specify the form the cowpea was offered (whole? chopped?...)

=> Ok, the cowpea offered was chopped before (L 130).

Line 107/Table 1: CF can be removed

=> Ok this has been done (L 133 – 134).

Line 110: DM as %DM is 100! No need to show DM

=> Ok this has been done (L 135).

Line 111: CA is not shown (‘Ash’ is enough)

=> Ok this has been deleted (L 136).

Line 114: were

=> Ok this has been done (L 139).

Line 115: removed and weighed

=> Ok this has been done (L 140).

Line 118: need to indicate times of day? consistent times?

=> Ok this has been done (L 145).

Line 121: need references

=> Ok this has been done (L 148 – 149).

Line 123: justify use of cows’ urine volume

=> Ok this has been done (L 154 – 151).

Line 160: loss

=> Ok this has been done (L 190).

Line 166: what about use of co-variates?

=> Ok, thanks for your comment, but we couldn't calculate the covariates and model clarified

 (L 196 – 202).

Line 182: are these final weights?

=> Ok, These are the average weights after 7 days of measurements. (L 218).

Line 184: breeds

=> Ok this has been done (L 220).

Line 204: not based on live weight

=> Ok this has been done (L 253).

Line 216: motility

=> Ok this has been done (L 267).

Line 218: based on what?

=> Ok, this part was reworded (L 263 – 269).

Reviewer 3 Report

Overall, this is a well written and good study.

I feel the discussion could do with being intertwined with more with previously published literature but is not bad overall. My only main concern is with the difference in live weights between the 2 treatments at the start of the experiment. If this difference is solely due to differences in mature weight between the breeds used it needs to be explained for readers unfamiliar with the breeds involved who are focused solely on the interesting energy and protein work.

Introduction:

Reference needed in first paragraph for first and final sentences (line 51 and 57).

Overall the introduction reads well but some more referencing throughout would be good.

Materials & Methods:

Line 96-97: How long before the experiment was Ivermectin administered? What vitamins were offered?

Line 114 - 116: Need to be reworded in clearer way

Results:

There was a significant difference in starting live weights for M & S breeds, Is this the reason why you are seeing the difference in dry matter consumption in M breed?

Discussion:

Line 215: I disagree with the use of 'close to a trend'. Something is either significant, trending or non-significant. Close to a trend is non-scientific and irrelevant.

Line 218: Where is the evidence to support this? If you believe this is the case in the data set then it needs to more carefully explained and discussed in this section.

Conclusion:

Needs to be written in a less speculative and rely more heavily on the findings from the paper.

Author Response

Animals

Overall, this is a well written and good study.

I feel the discussion could do with being intertwined with more with previously published literature but is not bad overall. My only main concern is with the difference in live weights between the 2 treatments at the start of the experiment. If this difference is solely due to differences in mature weight between the breeds used it needs to be explained for readers unfamiliar with the breeds involved who are focused solely on the interesting energy and protein work.

=> Ok this has been done

Introduction:

Reference needed in first paragraph for first and final sentences (line 51 and 57).

=> Ok this has been done (L53 and L58).

Overall the introduction reads well but some more referencing throughout would be good.

=> Ok this has been done

Materials & Methods:

Line 96-97: How long before the experiment was Ivermectin administered? What vitamins were offered?

=> Ok, One week before the adaptation period, all animals were dewormed with Ivermectin® 1%, received vitamins (stress vitam®)  (L 122 – 123).

Line 114 - 116: Need to be reworded in clearer way

=> Ok this has been done (L 139 – 140).

Results:

There was a significant difference in starting live weights for M & S breeds, Is this the reason why you are seeing the difference in dry matter consumption in M breed?

Ok, thank you for the question. I'll take no for an answer, because on an metabolic weight basis, DM intake differed between groups, showing that the voluntary DM intake is significantly higher in M.

Discussion:

Line 215: I disagree with the use of 'close to a trend'. Something is either significant, trending or non-significant. Close to a trend is non-scientific and irrelevant.

=> Ok this has been done (L 265).

Line 218: Where is the evidence to support this? If you believe this is the case in the data set then it needs to more carefully explained and discussed in this section.

=> Ok this has been done (L269).

Conclusion:

Needs to be written in a less speculative and rely more heavily on the findings from the paper.

 => Ok this has been done (L 305 – 312).

Reviewer 4 Report

Dear,

The experimental design is not appropriate because for any nutritional studies authors should have used balanced rations. Please explain how lactating goats were selected including their parities. Where is Majorera breed from? The appropriate title would be "Feeding Performance of Vigna unguiculata, variety 58/74 hay in Sahel and Majorera dairy goats".  Why did they not analyze this forage for mineral composition? I, therefore, recommend that this article be accepted as a research note or a short communication.

Notes:

Lie 184: replace race by breed

Line 194: two breeds

Line 305: M. a. Kahn should be replace by M. A. Khan

P-vales: Inconstantly spaced

Author Response

Animals

Dear,

The experimental design is not appropriate because for any nutritional studies authors should have used balanced rations. Please explain how lactating goats were selected including their parities. Where is Majorera breed from? The appropriate title would be "Feeding Performance of Vigna unguiculata, variety 58/74 hay in Sahel and Majorera dairy goats".  Why did they not analyze this forage for mineral composition? I, therefore, recommend that this article be accepted as a research note or a short communication.

 => Ok, thank you for your comments. We have tried to consider all. We have made significant changes in some parts of the document to better align the discussion with the results.

Notes:

Lie 184: replace race by breed

 => Ok this has been done (L 220).

Line 194: two breeds

 => Ok this has been done (L234).

Line 305: M. a. Kahn should be replace by M. A. Khan

 => Ok this has been done (L 423).

P-vales: Inconstantly spaced 

 => Ok this has been done  (L 212 – 213) and (L238).

Round 2

Reviewer 1 Report

Dear authors 

I am pleased to inform you that the article has been greatly improved in content and discussions.
It is satisfactory for the publication

Best regards

Reviewer 2 Report

Please check again spelling and grammar before final publication.

Reviewer 3 Report

All of my comments appear to have been dealt with satisfactorily. A thorough read through for English language checks is required though. An example of the errors that need to be corrected are in L63-63 where 'improve' is spelt incorrectly and the sentence does not flow correctly when reading.